# Prototyping for public health in a local context: a streamlined evaluation of a community-based weight management programme (Momenta), Northumberland, UK

Caroline J Dodd-Reynolds [1,2] Lisa Nevens,[3] Emily J Oliver,[1,2] Tracy Finch,[4] Amelia A Lake,[5,6] Coral L Hanson [7]

For numbered affiliations see end of article.

**Correspondence to**
Dr Caroline J Dodd-Reynolds; caroline.dodd-reynolds@durham.ac.uk

## ABSTRACT

**Objectives** Stakeholder co-production in design of public health programmes may reduce the 'implementation gap' but can be time-consuming and costly. Prototyping, iterative refining relevant to delivery context, offers a potential solution. This evaluation explored implementation and lessons learnt for a 12-week referral-based weight-management programme, 'Momenta', along with feasibility of an iterative prototyping evaluation framework.

**Design** Mixed methods evaluation: Qualitative implementation exploration with referrers and service users; preliminary analysis of anonymised quantitative service data (12 and 52 weeks).

**Setting** Two leisure centres in Northumberland, North East England.

**Participants** Individual interviews with referring professionals (n=5) and focus groups with service users (n=13). Individuals (n=182) referred by healthcare professionals (quantitative data).

**Interventions** Three 12-week programme iterations: Momenta (n=59), Momenta-Fitness membership (n=58) and Fitness membership only (n=65).

**Primary and secondary outcome measures** Primary outcome: Qualitative themes developed through stakeholder-engagement. Secondary outcomes included preliminary exploration of recruitment, uptake, retention, and changes in weight, body mass index, waist circumference and psychological well-being.

**Results** Service users reported positive experiences of Momenta. Implementation gaps were revealed around the referral process and practitioner knowledge. Prototyping enabled iterative refinements such as broadening inclusion criteria. Uptake and 12-week retention were higher for Momenta (84.7%, 45.8%) and Momenta-Fitness (93.1%, 60.3%) versus Fitness only (75.4%, 24.6%). Exploration of other preliminary outcomes (completers only) suggested potential for within-group weight loss and increased psychological well-being for Momenta and Momenta-Fitness at 12 weeks. 52 week follow-up data were limited (32%, 33% and 6% retention for those who started Momenta, Momenta-Fitness and Fitness, respectively) but suggested potential weight loss maintenance for Momenta-Fitness.

### Strengths and limitations of this study

- ► This study advances understanding about whether prototyping is a time-efficient and cost-effective approach to design and implementation of public health programmes.
- ► This mixed methods evaluation provides insight into the implementation of an 'off-the-shelf' weight management programme, in a local context.
- ► Embedding stakeholders' views throughout the entire evaluation process allowed for ongoing, iterative refinement.
- ► A limitation to the quantitative component is the small sample size and rate of missing data at 1 year; findings should thus be interpreted with caution.
- ► Qualitative interviews and focus groups can only provide information about what participants recall about their experiences, meaning that there is a potential for recall bias.

**Conclusions** Identification of issues within the referral process enabled real-time iterative refinement, while lessons learnt may be of value for local implementation of 'off-the-shelf' weight management packages more generally. Our preliminary data for completers suggest Momenta may have potential for weight loss, particularly when offered with a fitness membership.

## INTRODUCTION

Failure to implement effective public health interventions when programmes are scaled up or transferred across contexts is widely reported.[1] Proposed approaches attempting to address this implementation gap include: effectiveness-implementation hybrid designs,[2] linking action to theory and models based on theory[3] and application of the replicating effective programmes framework.[4] Common to all is advocacy of a developmental process reflecting on existing knowledge about the target population and planned programme

prior to service delivery. Furthermore, engagement of service users is encouraged at all stages of intervention and evaluation design in Medical Research Council guidance.[5] Although this increases the likelihood of services meeting all stakeholder needs, concerns about the practical, personal and professional costs of co-production have been raised.[6] Resulting well-designed services will be tailored to a problem that may have changed during the time spent developing the intervention. Additionally, public access may be delayed. Resource-pressured public health services must therefore consider pragmatic alternatives to service design and implementation. In this paper, we explore a novel evaluation approach to these implementation challenges, focusing on a problem high on the public health agenda: obesity and overweight.

Targeting elevated weight status is a public health priority, obesity being a recognised risk factor for many negative physical and psychological health outcomes.[7–11] In England for example, obesity and overweight are associated with 30 000 deaths and an estimated National Health Service cost of £6.1 billion per annum.[12] Globally, countries with higher income inequalities tend to have higher rates of obesity.[13] Excess weight is also associated with widening social and economic deprivation,[14] with calls to improve the effectiveness of behaviour change interventions for low-income groups.[15] There is a clear need for effective public health programmes that can be refined according to local need, especially in areas with substantial deprivation. This evaluation focuses on Northumberland, in North East England. Northumberland is one of the lowest ranked counties in England by Gross Value Added per capita (£16 140).[16] Unemployment is higher (5.5% vs 4.8%) than the England average[17], Northumberland public health spend per person is £53 compared with a £59 national average[18] and 63.8% of adults are classified as having excess weight, higher than the national average of 61.3%.[19]

The need for innovation within public health has been postulated, shifting away from the traditional linear preconceived and evidence-based model.[20] For example, Parry and colleagues[21] call for research to explore not only how a programme works, but also the context and requirements for any adaptations. One such approach is prototyping[22] where projects test innovations iteratively, with ongoing refinement considering the interplay between a programme and its delivery context. Evaluation and public health teams are able to communicate at all stages of the programme, with evaluation recommendations incorporated via a rapid-cycle basis.[21] A small number of studies to date, for example in drug prevention[22] and web-based support of long-term weight loss,[23] have demonstrated efficiencies when including elements of prototyping within programme development (including time, adaptation to context and cost). Such an approach seems particularly well-suited to weight management, where there are many examples of 'good' practice, or effectiveness, but no clear consensus on 'best' practice at service-delivery level. There is also

limited understanding of how 'scaling up' and adapting of programmes or interventions to local contexts may impact on effectiveness. This evaluation has particular value therefore in testing a prototyping approach for a weight management programme, delivered and adapted 'in real-time', at local authority level.

The aim was to explore implementation of an 'off-the-shelf' weight management programme, Momenta,[24] in a challenging context. Specific objectives were to explore local implementation, consider feasibility of the iterative prototyping evaluation framework and explore preliminary outcome domains including recruitment, retention, weight change and well-being.

## METHODS
### The prototyping process: local context and evaluation design
A local authority health needs assessment identified a gap in provision for a lifestyle-based weight management referral programme within Northumberland. Adults with overweight or obesity were at the time eligible for referral to the Northumberland exercise referral scheme (ERS), however previous evaluation demonstrated modest weight loss[25] and body mass index (BMI) $>30 \, kg/m^2$ was negatively associated with adherence.[26] Thus 'Momenta' was commissioned for local adaptation and delivery. The Momenta programme incorporates evidence-based behaviour change techniques and is designed to be delivered by fitness professionals in a leisure environment.[24] Developed by the MEND childhood weight management programme[27] designers, this 12-week programme aims to facilitate weight loss by engaging participants in 12 key behaviours broadly encompassing psychology, diet and physical activity (online supplementary file 1). Briefly, Momenta sessions explored topics using interactive and experiential learning techniques including brainstorming, group activities and discussion, quizzes and games. At the end of each session, participants set goals focusing on one of the 12 key behaviours. At the beginning of each session, the group discussed the previous weeks' goals by exchanging stories and brainstorming challenges. All interventions were free to service users.

The local Leisure Trust was commissioned to deliver a pilot Momenta programme. Commissioners and providers had ideas about alternative delivery options and due to an established academic relationship, asked the study team for advice about robust evaluation that would allow for feedback in real time and at the end of the pilot. Stakeholder meetings were held with public health staff (n=2), Leisure Trust managers (n=3), delivery staff (n=2) and Momenta programme developers (n=2). As part of the prototyping process, members of the evaluation team (CDR, EO) provided guidance on evaluation design and light touch advice about tools to explore preliminary effectiveness. The evaluation was thus co-produced to ensure a robust framework, while meeting strategic local needs. For example, commissioners were concerned about meeting recruitment targets for an existing specialist

weight management service used mainly for pre-bariatric patients and Momenta was initially commissioned for patients with BMI 25.0 to 29.9 kg/m$^2$, although this was later amended. Furthermore, commissioners were keen to consider accessibility of provision and wished to explore offering free gym, swimming and fitness class membership. The evaluation was designed to accommodate this.

The programme was ultimately delivered at two leisure sites situated within the 20% and 50% most deprived neighbourhoods in the country. Six general practice surgeries, identified as the best referrers to the existing ERS, were asked to refer suitable patients to Momenta. The programme manager and the public health improvement manager (LN) attended practice meetings to articulate referral criteria and disseminate advertising materials. Attendance varied from two to all practice staff, meaning that in some surgeries knowledge of the programme was reliant on dissemination by those who attended.

A mixed methods evaluation was agreed between the evaluation team and commissioners. Qualitative and quantitative components were conducted concurrently and had equal status.[28] Prototyping allowed for iterative changes to be made to the implementation and delivery of the programme in real time and we reflect on these in the results and discussion.

Referrals by healthcare professionals (HCPs) were via a standardised form to the appropriate leisure site. Prior to programme commencement the Leisure Trust, in conjunction with the Momenta programme designer and members of the evaluation team, held a training day for delivery staff. Although staff were qualified to deliver Momenta, extra bespoke training (including role-play scenarios and problem-solving discussions) was delivered by the clinical psychologist who designed Momenta. The evaluation team (CDR, CH) trained delivery staff in international standard anthropometric techniques[29] and familiarised them with other evaluation measures.

Programme providers allocated service users into one of three comparison groups: (a) Combined Momenta plus fitness membership (Momenta-Fitness), (b) Momenta and (c) Fitness membership (Fitness only).

Participants were allocated into groups in order of receipt (the first referral form received was allocated to Momenta-Fitness, the second form to Momenta, the third form to Fitness only, etc). The provider then contacted participants by telephone to arrange attendance. If a participant was unable to attend the allocated group, (e.g., due to inconvenient session times) the provider allocated them to a different group after discussion. Due to maximum recommended Momenta group size, referrals were split into delivery cohorts of 15, with groups rolling through March 2015 to April 2016.

Implementation effectiveness for the referral process was explored through semi-structured interviews with referring HCPs (undertaken at referring surgeries) and focus groups with service users (in leisure centres). All were conducted by LN during March 2015 to July 2015, as part of her public health master's degree (which contained qualitative methods training), mentored by TF, an experienced qualitative researcher. LN was employed as a member of the Northumberland public health team at the time of the evaluation. Questions are included in online supplementary file 2. Data were audio-recorded. Results are reported using the Consolidated criteria for Reporting Qualitative research guidelines.[30]

Practice managers from all six referring surgeries were sent an email invitation for staff to take part (n=84; General Practitioner=53, Practice Nurse=18, Healthcare Assistant=13). Individual correspondence was sent to those agreeing. LN informed participants about her employment status and that the study aimed to understand implementation issues. Interviews aimed to explore HCPs' referral experiences, raising weight issues, assessing readiness to change, marketing and referral materials, and the referral process. Interview questions were pilot tested with public health colleagues to assess timing and ensure validity. One question (*Thinking about after you referred the patient, what happened next?*) was omitted after piloting as it was realised HCPs would not have had patient feedback at that point. Interviews lasted on average 26 minutes and were transcribed verbatim. Data were analysed following each interview, with developing themes considered to determine whether questions required refinement. Initial themes generated from the first two interviews did not change and thus questions remained constant, although prompts were added.

During the initial assessment session for the first wave of referrals, all (n=39) were given a written invitation to participate in a series of focus groups at programme-end to explore experiences. Emphasis was placed on the referral process; initial expectations and experiences of participation; how weight issues were raised by HCPs; time from referral to initial assessment and facilitators and barriers to taking part. Focus groups lasted between 26 and 44 minutes.

Preliminary outcome data were collected to provide an initial indication of programme success. These included anthropometric measurements to determine weight change. Well-being measures were of specific interest to commissioners. Sociodemographic information was also available as indicated.

Age, gender and postcode (for index of multiple deprivation, IMD) were recorded by referring HCPs on the referral form. Employment status, level of education, cohort wave and programme group were recorded by leisure staff, who also measured weight and height (without shoes or bulky clothing) and waist circumference at baseline and programme end. Measures were taken in at least duplicate, using standardised tools in accordance with international standards[29] (SECA 761 scales, a Leicester portable stadiometer and anthropometry tape). Body mass index was calculated and classified according to WHO guidelines.[31] The Warwick-Edinburgh Mental Well-being Scale,[32] and the Hospital Anxiety and Depression Scale (HADS)[33] were administered at each time-point. Attendance at Momenta and leisure centre

usage were monitored via swipe-card tracking. Fifty-two weeks after commencing the programme, service users were invited to attend a follow-up session, where leisure staff repeated physiological and psychological measures. Programme providers collected and collated quantitative data and provided an anonymised data set to the evaluation team for analysis.

### Patient and public involvement

Patients and the public were not involved in the choice of evaluation topic, assisting in the study design, advising on the project or in carrying out the evaluation.

### Data analyses

Qualitative data were audio-recorded and transcribed by LN using a thematic process.[34] Data were organised according to concepts, key themes and developing categories. Data coding was discussed with TF, allowing comparison of data interpretation and subsequent coding refinement. Evolving key themes were refined through the analysis process and subsequent cross-sectional thematic labelling of data, thus generating deeper understanding. Where possible, key phrases or expressions identified from interviews and focus groups were retained within coding and thematic labelling. A public health colleague helped to verify interpretations of the data and appropriateness of codes applied. Once initial interviews were coded this framework was applied to remaining data. Notes taken during focus groups helped to contextualise when developing themes and included information about dynamics within groups, such as influence, disagreement, humour and peer exposure.

The anonymised quantitative data set was analysed using PASW Statistics V.22. Descriptive statistics were calculated for age, gender, IMD, employment status, initial BMI, leisure site, level of education and uptake and adherence. Distribution and normality of measures (weight, BMI, waist circumference, psychological well-being and attendance) were assessed using Shapiro-Wilk tests and median and IQR scores calculated for each group at baseline and 12 weeks (attendance, 12 weeks only). Using complete cases, Kruskal-Wallis H tests were used to explore preliminary between-group differences at baseline and at 12 weeks and Wilcoxon-signed rank tests explored preliminary repeated measures differences between baseline and 12-week scores. Complete cases available at 52 weeks (n=37) were considered similarly, but via separate analyses due to limited available data across comparison groups.

### RESULTS

Between December 2014 and March 2016, the programme received 182 referrals and was delivered in four cohorts across leisure sites. Due to initial low levels of recruitment, the first cohort did not start until March 2015. Referrals were mainly female and 27.5% lived in the 20% most deprived areas (table 1).

**Table 1** Demographic characteristics of referrals who started the programme (n=153)

|  | Median | IQR |
|---|---|---|
| Age (years) | 53 | 24 |
| Gender | **n** | **%** |
| Male | 25 | 16.3% |
| Female | 120 | 78.4% |
| Not stated | 8 | 5.2% |
| Initial BMI category (kg/m$^2$) | | |
| 25.0–29.9 | 40 | 26.1% |
| 30.0–34.9 | 73 | 47.7% |
| 35.0–39.9 | 27 | 17.6% |
| ≥40.0 | 10 | 6.5% |
| Not stated | 3 | 2.0% |
| Leisure site | | |
| Leisure site 1 (IMD quintile 2) | 69 | 45.1% |
| Leisure site 2 (IMD quintile 3) | 83 | 54.2% |
| Not stated | 1 | 0.7% |
| Index of multiple deprivation | | |
| 20% most deprived | 42 | 27.5% |
| 21%–40% | 33 | 21.6% |
| 41%–60% | 17 | 11.1% |
| 61%–80% | 20 | 13.1% |
| 20% least deprived | 35 | 22.9% |
| Not stated | 6 | 3.9% |
| Employment status | | |
| Employed full-time | 36 | 23.5% |
| Employed part-time | 24 | 15.7% |
| Retired | 51 | 33.3% |
| Claiming incapacity benefit | 5 | 3.3% |
| Claiming job seekers allowance | 6 | 3.9% |
| Not stated | 14 | 9.2% |
| Level of education | | |
| Primary | 15 | 9.8% |
| Secondary (O level/GCSE) | 35 | 22.9% |
| Secondary (A level) | 26 | 17.0% |
| Further education (HND) | 24 | 15.7% |
| Bachelors or equivalent | 21 | 13.7% |
| Masters or equivalent | 5 | 3.3% |
| Not stated | 27 | 17.6% |

Age, gender and postcode (IMD calculated by the programme provider) recorded from the referral form.
BMI and leisure site recorded by the provider. Missing data not available for analysis and presumed to be data entry errors.
Employment and level of education self-reported by participants during the first session. The provider did not follow-up missing data.
BMI, body mass index; GCSE, General Certificate of Secondary Education; HND, Higher National Diploma; IMD, index of multiple deprivation.

## Implementation effectiveness: reflections from referring healthcare professionals

Five face-to-face semi-structured interviews took place with HCPs across five referring surgeries: two general practitioners (GPs), two Practice Nurses and one Healthcare Assistant. HCPs perceived that four key themes influenced the effectiveness of programme implementation: (i) difficulties raising weight with patients, (ii) how gender affected patient engagement, (iii) availability of information and resources and (iv) additional barriers constraining programme promotion.

### Raising the issue of weight with patients:

Concerns about raising weight may have contributed to slow recruitment, with nurses and healthcare assistants expressing unease, *'not really up to me… well I talk about it if they want to…. Better if they (patients) bring it up.'* (Interview 2, Healthcare Assistant). GPs seemed more comfortable raising weight with patients, but with the caveat that this is easier in the context of a longer-term GP/patient relationship.

> 'the people I see I've known for a very long time… it's the rapport you have…if I'd never met anyone before and they came in for a sore throat I'm not going to say you're fat…If there was someone I'd known for a long time and it seemed relevant…I'd mention it.' (Interview 5, GP).

### Gender and engagement in the referral process:

Gender was highlighted as influencing the referral process, with women being more likely than men to seek support for weight. This may help explain the low rate of referral for males (17%):

> 'More women talk about it…men don't really talk about weight…I do mention weight to men if I'm doing a well man (sic) but it doesn't come up really… it's a woman thing…' (Interview 3, Practice Nurse).

### Availability of information and resources:

Several interviewees highlighted training needs around programme information and resources, (e.g., additional programme information would help to engage patients). For example, the GPs both discussed the longstanding ERS and stated they needed to become more familiar with Momenta, as they had with the ERS:

> 'when we get opportunities to do things in the practice we normally discuss it, let everyone know where appropriate forms and information is and it's in your head…that didn't happen with this and I don't know why that was.' (Interview 5, GP).

All HCPs interviewed felt the referral leaflet (provided by programme providers) was important in the process, either as a tool to promote the intervention or to convey information to patients:

> 'The leaflet was good, bright…explained the programme and patients like taking a leaflet away.' (Interview 3, Practice Nurse).

### Additional barriers to engagement:

Several subthemes highlighted additional barriers to the referral process. The most prominent were around initial BMI referral criteria (25.0 to 29.9 kg/m$^2$) and delayed programme start. Both implementation factors were beyond the control of the referrers, but consequently amended through iterative refinement during the prototyping process following early data analysis. Both were reported by practice nurses as exacerbating each other:

> 'we were referring but then it didn't start so people were not sure what was happening (pause)…Think it was more people were needed to start…but you know if the BMI was higher then there would have been more.' (Interview 3, Practice Nurse).

In one case, a decision was taken to relax the referral criteria, *'…31.5 (kg/m$^2$)…was just outside so I just referred him.'* (Interview 4, GP).

Programme location was perceived by HCPs to overcome an existing barrier to the tier three weight management programme, as Momenta was *'round the corner for people,'* as opposed to *'a bit far away at the hospital.'* Cost barriers were also discussed, both with reference to the patient, *'in this sort of area…cost…, if you've got to pay it's a barrier.'* (Interview 4, GP), and to expected targets from Clinical Commissioning Groups (CCG),

> 'we are constantly told by the CCG that we must keep down on numbers and that if there are costs attached to this referral that would definitely impact… and that would be for all practices.' (Interview 5, GP).

## Implementation effectiveness: reflections from participants

Three focus groups in the leisure centres allowed programme participant voices to be heard: three females and one male from Momenta (focus group 1), three males and three females from Momenta-Fitness (focus group 2) and three females (one of whom emailed her views separately) from Fitness only. Across the groups, 12 participants reported having lost weight and one reported weight gain. Three themes developed: (i) outcomes of the programme, (ii) facilitators and barriers to engagement and (iii) raising the issues of weight with HCPs.

### Outcomes of the programme:

Focus group findings aligned closely with quantitative outcomes in terms of the physical and psychological benefits of participation: *'(I've) lost a good bit of weight. It's been very positive for me… I'm feeling a lot more active…'* (Momenta-Fitness, Participant 5). Participants reported a sense of weight loss achievement, increased physical activity levels and positive mood states. In addition, elements of the Momenta programme were perceived as facilitating engagement, including the *'group feeling… I looked forward*

to it,' (Momenta-Fitness, Participant 4), the *'information that we got every week… so very well planned.'* (Momenta-Fitness, Participant 3) and the ongoing support for example, *'she 'phoned me the other day to see if I was coming,'* (Momenta-Fitness, Participant 4). Momenta participants reflected back on, and identified and discussed lifestyle factors that related to their initial weight gain (eg, *'I did the usual thing… I started eating toffees,'* Momenta-Fitness, Participant 5), demonstrating both self-awareness and an openness to discussing the topic.

### Facilitators and barriers to engagement:

One participant reported being initially excluded but later allowed to take part, and others raised concerns that the initial BMI threshold for referral (25 to 29.9 kg/m$^2$) was too low, *'was a little bit high, BMI…managed to get it down… (and then) the doctor put us forward,'* (Momenta, Participant 2). Data also indicated the importance of subsidised access, particularly important in the context of a deprived region such as this, for example, *'I also joined Weight Watchers for short period of time but found the classes too expensive,'* (Fitness only, Participant 3, emailed response).

### Raising the issue of weight with HCPs:

Some data did suggest implementation was problematic, however this focused exclusively on the referral process. Participants overwhelmingly felt that they had opened the conversation about weight, as opposed to discussions being initiated by HCPs (eg, *'my glucose levels were quite high but nobody ever said that I was overweight,'* Momenta-Fitness, Participant 4). In addition, participants perceived limitations in HCPs' knowledge of intervention components (*'she (nurse) didn't know anything about it,'* Fitness only, Participant 1), something with potential to impact on likelihood of referral, and participants' expectations of programme success.

### Preliminary outcome domains

Of all referrals, 153 (84%) attended the baseline measurement session and 78 (51% of those who started) attended the 12-week measurement session. Uptake and adherence varied by programme group (table 2).

Physiological and psychological data were not normally distributed. Quantitative data are presented as exploratory, due to the small sample size and are presented here for information and general description. No differences were found between programme groups either at baseline or at 12 weeks, for any measures. Despite the small sample size, within-group changes between baseline and 12 weeks were evident for weight, BMI and waist circumference for Momenta-Fitness, and Momenta (table 3). Follow-up analysis at 52 weeks (available subsample)

| Table 2 | Programme uptake, adherence and attendance | | | |
|---|---|---|---|---|
| **Uptake (week 1), adherence and retention (both week 12)** | | **Momenta-Fitness** | **Momenta only** | **Fitness only** |
| Number referred | | 58 | 59 | 65 |
| Uptake* (n, %) | | 54 (93.1%) | 50 (84.7%) | 49 (75.4%) |
| Uptake retention† (n, %) | | 35 (64.8%) | 27 (54.0%) | 16 (32.7%) |
| Uptake adherence‡ (n, %) | | 34 (63.0%) | 26 (52.0%) | 8 (50.0%) |
| Overall retention§ (n, %) | | 35 (60.3%) | 27 (45.8%) | 16 (24.6%) |
| Overall adherence¶ (n, %) | | 34 (58.6%) | 26 (44.1%) | 8 (12.3%) |

| | **Momenta-Fitness** | | **Momenta only** | | **Fitness only** |
|---|---|---|---|---|---|
| **Momenta session attendance** | n | Median (IQR) | n | Median (IQR) | |
| Uptake | 54 | 9.0 (7.3) | 50 | 9.0 (8.0) | |
| Dropouts | 19 | 3.0 (3.0) | 23 | 3.0 (5.0) | N/A |
| Completers** | 35 | 10.0 (2.0) | 27 | 11.0 (1.3) | |

| | **Momenta-Fitness** | | **Momenta only** | | **Fitness only** | |
|---|---|---|---|---|---|---|
| **Exercise session attendance** | n | Median (IQR) | n | Median (IQR) | n | Median (IQR) |
| Uptake | 54 | 7.0 (16.3) | 50 | 0.0 (4.5) | 49 | 0.0 (1.5) |
| Dropouts | 19 | 0.0 (1.0) | 23 | 0.0 (0.0) | 33 | 0.0 (0.0) |
| Completers** | 35 | 10.0 (14.0) | 26 | 0.0 (5.0) | 16 | 4.5 (18.0) |

*Uptake, participant attended baseline assessment;

†Uptake retention, % of participants who attended the 12 week assessment out of those who attended the baseline assessment;

‡Uptake adherence, % of participants who attended the baseline assessment who also attended ≥eight Momenta sessions (Momenta-Fitness and Momenta only) or gym sessions (Fitness only);

§Overall retention, % of all those referred who attended both baseline and 12 week assessment;

¶Overall adherence, % of all those referred who attended ≥eight Momenta sessions (Momenta-Fitness and Momenta only) or exercise sessions (Fitness only);

**Completers, those who completed the 12 week assessment.

**Table 3** Weight, body mass index and waist circumference change

| End of programme results | Median (IQR) Baseline | Median (IQR) 12 weeks | z | Median (IQR) Change |
|---|---|---|---|---|
| **Weight (kg)** | | | | |
| Momenta-Fitness (n=35) | 88.9 (80.5–100.0) | 88.0 (77.2–95.8) | −4.531 | −2.9 (-5.1–-1.6) |
| Momenta only (n=26) | 87.8 (74.5–77.0) | 83.3 (74.5–92.5) | −4.344 | −2.9 (-5.0–-2.0) |
| Fitness only (n=15) | 76.2 (71.6–86.9) | 76.6 (70.4–84.6) | −0.879 | 0.0 (-3.2–1.0) |
| **BMI (kg/m$^2$)** | | | | |
| Momenta-Fitness (n=35) | 32.0 (30.3–35.7) | 31.3 (29.2–35.3) | −4–494 | −1.1 (-1.9–-0.6) |
| Momenta only (n=26) | 32.0 (30.0–34.5) | 31.3 (28.6–33.6) | −4.356 | −1.2 (-1.6–-0.8) |
| Fitness only (n=14) | 29.2 (27.3–33.0) | 29.7 (27.0–33.3) | −0.454 | 0.1 (−1.2–0.4) |
| **Waist circumference (cm)** | | | | |
| Momenta-fitness (n=35) | 106.0 (98.0–115.0) | 99.0 (93.0–110.0) | −4.996 | −7.0 (-9.5–-5.0) |
| Momenta only (n=25) | 108.0 (99.5–114.5) | 101.0 (93.8–111.5) | −4.166 | −5.0 (-7.3–-2.5) |
| Fitness only (n=11) | 90.0 (87.0–95.0) | 91.0 (90.0–96.0) | 0.358 | 1.0 (-3.0–3.0) |
| **52 week follow-up** | Median (IQR) Baseline | Median (IQR) 52 weeks | z | Median (IQR) Change |
| **Weight (kg)** | | | | |
| Momenta-Fitness (n=18) | 95.2 (87.1–101.4) | 91.4 (82.7–95.9) | −3.006 | −4.8 (-6.2–-1.5) |
| Momenta only (n=16) | 84.7 (72.3–95.2) | 82.7 (73.2–94.6) | −1.533 | −0.7 (-7.6–0.8) |
| Fitness only* (n=3) | 73.4 (69.5–80.2) | 70.3 (66.0–87.0) | | 0.9 (-7.4–6.9) |
| **BMI (kg/m$^2$)** | | | | |
| Momenta-Fitness (n=18) | 32.0 (30.49–35.1) | 30.8 (28.7–34.0) | −3.157 | −1.7 (-2.0–-0.6) |
| Momenta only (n=16) | 31.7 (29.3–33.9) | 31.1 (26.7–33.6) | −1.603 | −0.3 (-2.3–0.3) |
| Fitness only* (n=3) | 27.6 (27.5–30.5) | 27.8 (24.8–33.2) | | 0.3 (24.8–33.2) |
| **Waist circumference (cm)** | | | | |
| Momenta-Fitness (n=18) | 109.0 (101.0–114.8) | 100.5 (94.8–107.3) | −3.221 | −6.0 (-13.3–-1.75) |
| Momenta only (n=16) | 106.0 (94.5–115.8) | 103.5 (98.5–113.3) | −0.780 | −2.5 (-9.0–10.0) |
| *Fitness only (n=3) | 89.0 (87.0–95.0) | 90.0 (90.0–101.0) | | 3.0 (90.0–101.0) |

*Fitness only n=3 therefore median and range reported and no statistical test completed.

suggested changes were maintained for Momenta-Fitness (n=18) only.

Differences in mental well-being, depression and anxiety were not apparent between groups, however improvements in mental well-being, and reductions in depression and anxiety were suggested between baseline and 12 weeks for Momenta-Fitness and Momenta groups only (table 4), although the magnitude of change was similar for all groups. Subsample analysis at 52 weeks demonstrated potential for improvements for well-being and depression to be maintained for Momenta-Fitness, and well-being and anxiety for Momenta.

Overall, the results suggested those who participated in the two groups incorporating Momenta, had enhanced physical and psychological health indicators from baseline, whereas those who had only free fitness membership did not. From the small follow-up sample, there is scope to suggest that the combination of Momenta and fitness membership may produce favourable outcomes at 52 weeks.

### Iterative refinements throughout the evaluation process

Here we list a number of implementation adjustments which were made throughout the evaluation process, facilitated via the prototyping framework. Real-time advice from commissioners was considered during early stages of implementation, regarding the nature of comparison offers (e.g., fitness access) and thus initial design and outcome measurements were adapted prior to referrals being made. To better-target recruitment and change the process of engagement at referral point, entry criteria were altered to also include BMI ≥30 kg/m$^2$ mid-way through programme delivery. On-site implementation of the service offer was adapted in response to delivery staff feedback: increased resource was made available, for example additional staffing to support delivery for the first wave of referrals. Furthermore, staff were given

**Table 4**  Well-being, anxiety and depression measures change

| End of programme results | Median (IQR) Baseline | Median (IQR) 12 weeks | z | Median (IQR) Change |
|---|---|---|---|---|
| **Mental well-being scale** | | | | |
| Momenta-Fitness (n=29) | 46.0 (40.0–51.5) | 53.0 (40.0–51.5) | 3.810 | 5.0 (1.5–12.0) |
| Momenta only (n=23) | 49.0 (39.0–58.0) | 55.0 (51.0–63.0) | 2.818 | 6.0 (-1.0–10.0) |
| Fitness only (n=13) | 47.0 (40.5–59.5) | 46.0 (42.0–63.5) | 0.157 | 0.0 (-4.0–5.0) |
| **Anxiety scale** | | | | |
| Momenta-Fitness (n=28) | 5.5 (4.0–9.8) | 4.5 (2.0–7.0) | −3.027 | −1.0 (-3.0–1.0) |
| Momenta only (n=23) | 8.0 (6.0–10.0) | 4.0 (2.5–9.0) | −2.329 | −1.0 (-3.0–0.0) |
| Fitness only (n=13) | 8.0 (3.5–10.0) | 6.0 (4.0–9.0) | −0.499 | −1.0 (-2.0–2.0) |
| **Depression scale** | | | | |
| Momenta-Fitness (n=28) | 5.5 (3.3–8.0) | 2.0 (1.0–6.0) | −3.214 | −2.5 (-4.8–-0.3) |
| Momenta only (n=23) | 5.0 (3.0–7.5) | 3.0 (1.0–5.0) | −3.049 | −1.0 (-4.5–1.0) |
| Fitness only (n=13) | 4.0 (2.0–8.5) | 2.0 (2.0–7.0) | −1.226 | −2.0 (-4.5–0.0) |

| 52 week follow-up | Median (IQR) Baseline | Median (IQR) 52 weeks | z | Median (IQR) Change |
|---|---|---|---|---|
| **Mental well-being scale** | | | | |
| Momenta-Fitness (n=15) | 44.0 (39.0–52.0) | 55.0 (48.0–59.0) | 2.984 | 5.0 (3.0–15.0) |
| Momenta only (n=13) | 58.0 (47.5–59.0) | 56.0 (54.0–63.5) | 2.282 | 4.0 (0.5–6.5) |
| Fitness only* (n=3) | 47.0 (34.0–64.0) | 58.0 (45.0–60.0) | | −2.0 (-6.0–26.0) |
| **Anxiety scale** | | | | |
| Momenta-Fitness (n=15) | 6.0 (2.0–10.0) | 2.0 (1.0–7.0) | −1.785 | −3.0 (-6.0–0.0) |
| Momenta only (n=15) | 7.0 (4.0–9.0) | 5.0 (1.0–8.0) | −1.990 | −3.0 (-4.0–0.0) |
| Fitness only* (n=3) | 9.0 (5.0–10.0) | 2.0 (1.0–8.0) | | −3.0 (-8.00–-2.0) |
| **Depression scale** | | | | |
| Momenta-Fitness (n=15) | 7.0 (3.3–11.3) | 3.5 (1.0–6.0) | −2.908 | −3.5 (-6.3–-0.8) |
| Momenta only (n=15) | 4.0 (1.0–6.0) | 3.0 (1.0–4.0) | −0.762 | 0.0 (-2.0–1.0) |
| *Fitness only (n=3) | 3.0 (0.0–8.0) | 2.0 (1.0–8.0) | | 1.0 (-8.0–5.0) |

*Fitness only n=3 therefore median and range reported and no statistical test completed.

additional time for Momenta session preparation and session delivery times were extended. Follow-up activities (i.e., text or telephone contact) were implemented by staff during the process, to encourage adherence.

## DISCUSSION

We explored 'prototyping', as a cost-effective and time-efficient approach to public health evaluation, via an 'off-the-shelf' weight management programme implemented in a local context of mixed and high deprivation.

Experiences of both referrers and referrals highlighted that HCPs needed to be better-informed and more confident raising weight-related conversations. While patient-led action is desirable, staff reluctance to raise weight issues may mean that opportunities for engagement of less knowledgeable or motivated patients will be missed. The problematic positioning of GPs within obesity care has been highlighted previously,[35] with a range of strategies to change HCPs' behaviour resulting

in little or no change to patients' weight. A practical training need is highlighted for those working at the patient-practitioner interface, however communication with patients about weight may well be hindered by the 'stigma' attached to obesity.[36] This has wider implications for patient outcomes and requires further exploration through the implementation process. Additionally, HCPs need better understanding of referral-based public health programmes offered. Despite efforts of programme and public health managers, awareness was reportedly low for some referring professionals. We suggest consideration of resource-efficient ways to signpost both HCPs and patients themselves as part of the implementation process.

This programme was delivered across a social gradient in a region with low health indices and areas of high deprivation. Some issues in relation to inequalities and service access for future community-based weight management programmes were highlighted. Only 16% of referrals to Momenta were males. Gender bias in weight management

referral has been reported elsewhere,[37 38] and interviews showed that practitioners struggled to raise the topic of weight with male patients. Alternative referral strategies have been employed in other settings in an attempt to overcome this.[39] Marketing in other community spaces, or targeted postal referrals could be explored in future implementation. The initial decision to restrict referral to overweight-only substantially impacted on referral rates, with HCPs and referrals indicating they felt limited until this restriction was reversed. Had this continued, worsening health inequalities may have been an unintended consequence, something to be actively avoided within public health programmes.[40] The roles of, and interactions between, those operating in the 'system' (i.e., the context within which the intervention operates) must be considered at the point of implementation to minimise any impact from unintended consequences.[5] In practical terms, this may be through continued dialogue with commissioners, referring professionals and referrals themselves, something which prototyping evaluation allows.

Quantitative data should be interpreted as exploratory, due to the relatively small number of complete cases, however lessons can be learnt from these data both in terms of preliminary outcomes and engagement/dropout. Participation in Momenta and Momenta-Fitness resulted in 12-week weight loss for those who completed the programme. Free fitness membership without the weight-management programme was poorly engaged with and did not lead to weight change. A small subsample who attended follow-up demonstrated that after 1 year, weight reductions equivalent to ~4% could be maintained for Momenta-Fitness. We caution that while this might be best interpreted as hypothesis-generating for future evaluations, given these effects emerged despite an underpowered sample it is worth briefly considering potential mechanisms here. Providing free access to fitness facilities alongside the behaviour change programme may allow for continuous and self-driven behaviour change[41] and sustaining optimal changes in adiposity over 12 months in those who remained engaged.[42] Swipe card monitoring during the initial 12-week period indicated that fitness sessions were accessed an average of 10 occasions for this group, whereas no access was apparent for Momenta, despite Momenta sessions being held in leisure centres. This could be important for community providers making decisions about delivery location. Both Momenta groups reported improved well-being, and reduced anxiety and depression at 12 weeks. The changes observed, though small, could be argued to approach being functionally and clinically meaningful, with a minimal important difference of 1.5 points previously identified for the HADS, for example.[43] The behavioural intervention may drive this effect. This is consistent with previous work reporting covarying changes in weight loss, depression and quality of life in weight management services.[44] It is unclear whether the primary mechanism was weight loss, or the wider social benefits of participation. Both were valued in

the qualitative data. Our preliminary evidence of maintained improvements in well-being for these groups at 52 weeks is particularly relevant given previously evidenced associations between poor mental health, and obesity and overweight status.[45] Long-term follow-up rates will need to be considered in future similar programmes and we suggest year-long follow-up (at least) is included as a key programme component from the outset. Consideration should be given to how providers can maintain contact with participants after programme end to increase likelihood of successful follow-up. Potential 'light touch' support after 12 weeks may be helpful and other means of obtaining follow-up data should be explored where service users disengage. Reasons for disengagement might also be usefully explored in future work.

Given that no systematic problems emerged with service-users' experiences of the programme itself, our findings lend support to a streamlined approach to involvement of all stakeholders in programme implementation. We suggest that prototyping demonstrates opportunities for off-the-shelf programmes to be pragmatically moulded to local contexts, in real-time. Many of the iterative changes made were staff-driven. This demonstrates that real-time consideration of feedback from on-site delivery teams can be important to the implementation process. Some of the adjustments required commissioning action, as they had resource implications; others needed advice from the evaluation team. Interestingly changes made throughout the process generally focused on both staff and participant experience.

Emergence of some negative experiences of referral suggests, however, that prototyping can be problematic without networks or channels for ensuring key outcomes are widely communicated to relevant stakeholders. Overall, the evaluation demonstrated that a balance is needed to allow quick and efficient adaptation of off-the-shelf programmes, but with focused professional user engagement in the early stages of development. The prototyping approach had particular utility given that project resources were limited and meant that issues were identified and acted on rapidly. While the programme may have progressed similarly without this, prototyping provided a greater structure for, and confidence in, on-going refinements. This was achieved via the support provided by academics, public health practitioners and providers. Fundamentally, adopting a prototyping approach enabled the delivery of a new service to an in-need population, alongside the generation of initial evidence of local effectiveness. A minimum of 1 kg weight-loss at 3 months, and 0.7 kg at 12 months have been suggested as thresholds to influence decisions over commissioning of weight-loss services.[46] Our preliminary data suggests that Momenta may have potential to meet or even exceed these thresholds, showing particular promise when implemented in conjunction with free fitness provision.

Demonstrating preliminary effectiveness is of limited use, however, unless a successful programme in one area

may be adapted and implemented to suit a different context, for example through sharing local-level knowledge, interactions and behaviours of individuals within different parts of that system.[47] The process for scaling-up of effective health interventions to broader policy and practice takes years[48] and certainly within the obesity literature, has been dominated by initiatives that consider effectiveness but not implementation across specific settings.[49 50] We recommend prototyping might be built into larger public health evaluations providing that the original programme has a sound theoretical basis, and iterative refinement is engaged with by all stakeholders from the outset.

## CONCLUSION

The Momenta programme was experienced positively by those who attended. Issues with the referral process need to be explored further, however other refinements were feasible during delivery. Promising preliminary outcome data for completers of 'Momenta', particularly in conjunction with a free fitness offer, implies potential for the scheme within future commissioning. This evaluation extends the literature by exploring prototyping for a complex problem, community weight-management, in a challenging setting, demonstrating streamlined implementation of an 'off-the-shelf' weight management programme. This resource-effective approach is highly relevant in the context of health inequalities and public health sector funding constraints.

**Author affiliations**
[1]Department of Sport and Exercise Sciences, Durham University, Durham, UK
[2]Wolfson Research Institute for Health and Wellbeing Physical Activity Special Interest Group, Durham University, Durham, UK
[3]Northumbria Healthcare NHS Foundation Trust, Northumberland, UK
[4]Department of Nursing, Midwifery and Health, Northumbria University, Newcastle upon Tyne, UK
[5]Department of Science, School of Science, Engineering and Design, Teesside University, Teesside, UK
[6]Fuse – UKCRC Centre for Translational Research in Public Health, North East England, UK
[7]School of Health and Social Care, Edinburgh Napier University, Edinburgh, UK

**Acknowledgements** Many thanks to Jordan Bell, Nicole Rowley and Ross Davison, who collected and collated raw quantitative data.

**Contributors** CDR, CH and EO contributed to design of the quantitative evaluation, data analysis and preparation of the final document. LN contributed to design and analyses of the qualitative evaluation component. TF contributed to qualitative design and analysis and preparation of the final document. AL contributed to design of the quantitative evaluation and preparation of the final document.

**Funding** The quantitative evaluation component was supported by funding from the Wolfson Research Institute for Health and Wellbeing, Durham University. Amelia Lake is a member of Fuse, the Centre for Translational Research in Public Health ( www.fuse.ac.uk). Fuse is a UK Clinical Research Collaboration (UKCRC) Public Health Research Centre of Excellence. Funding for Fuse from the British Heart Foundation, Cancer Research UK, Economic and Social Research Council, Medical Research Council, the National Institute for Health Research, under the auspices of the UKCRC, is gratefully acknowledged.

**Competing interests** CH is a former employee of the Leisure Trust and was instrumental in initiating the evaluation. She was subsequently employed as a research assistant at Durham University, however was not involved in any data collection or entry, only accessing an anonymised database submitted to the

University. LN was a Public Health Improvement Manager within the Public Health Team and had responsibility for commissioning the Momenta programme. The qualitative evaluation component was submitted in partial fulfilment of her Master's in Public Health at Newcastle University.

**Patient consent for publication** Not required.

**Ethics approval** Ethical advice was sought from the local Research Manager of North of England Commissioning Support, and this project was classed as a service evaluation in line with National Research Ethics Service guidance.

**Provenance and peer review** Not commissioned; externally peer reviewed.

**Data availability statement** No data are available.

**ORCID iDs**
Caroline J Dodd-Reynolds http://orcid.org/0000-0003-0670-8264
Coral L Hanson http://orcid.org/0000-0003-1602-1968

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
