## [Reviewer comments · BMJ Open]

ARTICLE DETAILS

TITLE (PROVISIONAL)	Prototyping for public health in a local context: a streamlined evaluation of a community-based weight management programme (Momenta), Northumberland, UK
AUTHORS	Dodd-Reynolds, Caroline; Nevens, Lisa; Oliver, Emily; Finch, Tracy; Lake, Amelia; Hanson, Coral

VERSION 1 – REVIEW

REVIEWER	Dimitrios Koutoukidis University of Oxford UK
REVIEW RETURNED	20-Mar-2019

GENERAL COMMENTS	Abstract Overall This is a useful study showing the challenges of scaling up weight loss programmes in routine care. Major comments Given the 20% retention rate at 1-year and the lack of accounting for missing data, authors should be more cautious in the interpretation of the effectiveness of the programme and this should be reflected both in the abstract, results, and discussion. I suggest that the abstract is overall framed more towards “lessons learned” rather than effectiveness. Methods Page 11, line 16: it is unclear what does viable data mean. Is these complete cases? If so, why were the missing data ignored and what might be the implication of this? Although it is perfectly reasonable to report on completers, the resulting effect sizes are likely to be largely inflated by those who mostly adhered to the programme. Authors should also report effects on weight (and other measures) using multiple imputation (or full information maximum likelihood) at 12-weeks, given the very high dropout rate (which is typical in weight loss programmes). Results Authors should consider evaluating adherence though session attendance rather than attendance at 12-week follow-up. The latter evaluates retention rather than adherence. Please revise accordingly. Minor comments
--

	Please add the weight data for fitness only Please rephrase 'persistence of weight loss' to 'weight loss maintenance' and provide numerical data for all 3 conditions. Line 7: Correct grammatical error Line 8: Change "a" to lower case Introduction Line 12: Correct error "efficiencies" Methods Line 7 and throughout the manuscript: Please use person-first language when referring to people with overweight/obesity. Line 8: unclear what "doubled" means Line 9: delete "a" Page 8, line 5: country or county? Page 8, line 17: Please expand on how the allocation was implemented. Was it random, did people have a choice, etc. Page 11, line 18: Could the authors elaborate on the manuscript on why two different models were chosen instead of one model for all time points? Page 21, line 4: does this refer to the upper BMI threshold? Discussion Page 22, line 3: not sure where in the manuscript the 5% wl is reported? Missing data should be accounted for before comparison with other programmes can be made. The authors argue that the "the prototyping evaluation format allowed for changes following programme commencement". These changes would be really interesting and useful for the readers and I suggest you incorporate them in the results. Key points should be less abstract and more close to the data.
--	---

REVIEWER	Stephan Dombrowski University of New Brunswick, Canada
REVIEW RETURNED	17-Apr-2019

GENERAL COMMENTS	Thank you for the opportunity to review this manuscript. It presents an interesting study examining the implementation of an "off the shelf" weight management programem within a local context using a mixed methods design. Although the study examines a pressing issue in obesity and implementation, there are several comments that could be made on the current version of the manuscript. There appears to be a disconnect between the objectives of the manuscript and large portions of the presented results. This is a study that is presented as examining prototyping, but the results focus primarily on what are probably underpowered weight loss and other secondary outcomes. The current study is unlikely to be powered to provide an evaluation of effectiveness, the title might need amending and some of the presented results need to be interpreted with caution. Momenta is presented as "evidence based" throughout the manuscript, but it is unclear if this programme has been developed based on evidence, or if there is evidence of its effectiveness in inducing weight change? Please clarify.
--

	Statements such as “Engagement of service users is encouraged”, p.5 need referencing – encouraged by whom? The cost argument against user engagement in the introduction needs to be supported by evidence. It is not entirely clear why this would be relevant to make the case for the current study. Such statements might just antagonise some readers. More detail on what is specifically meant by ‘prototyping’ would be useful and a review of background evidence would add value in the introduction. How would this differ from feasibility and pilot studies? There might be other study designs (e.g. open pilots) which are not called ‘prototyping’, but are similar to what has been attempted here could be examined to provide background information. How is the adaptability of the programme (mentioned on p. 6) determined in the context of the current study? Please consider ending the introduction section with a clear statement of aim and objectives. “Tier three weight management service” (p.7) might not be universally understandable to readers. More details on how the “prototyping approach” was determined and implemented would add value. More information on the specific design choices would be useful. How were service users allocated to the three groups (p.8)? If not at random this needs to be clearly stated. The number of study staff attending study meetings needs a denominator (p.8), or present as percentage to preserve anonymity. The number of HCPs and practice managers need to be added to the methods, p.10. Please provide details on how exactly anthropometric outcomes were collected. Please provide a brief description of the content of the interventions groups. The patient and public involvement section states that “Commissioners, deliverers and service users were involved in the iterative evaluation.” But weren’t these subject of the investigation, rather than involved in the research study? Please clarify and provide more details if I misinterpreted this. The results focus on outcomes mainly, when the aim of the article seems to be to examine implementation using the prototyping approach. There seems to be a disconnect between the general introduction, and the presented results. More emphasis should be given to dropout, which is high, as is not uncommon in these types of studies and programmes. Given the small sample size, no imputation methods being used, and the lack of randomisation (I assume), the interpretation of the findings should be more cautious.
--	---

	One could argue that outcomes should be presented in line with recommendations for the CONSORT extension for pilot and feasibility trials (which this study resembles), e.g. removing p-values. It might be beneficial if the entire manuscript followed the CONSORT checklist, see http://www.consort-statement.org/extensions/overview/pilotandfeasibility “the prototyping evaluation format allowed for changes following programme commencement, suggesting that this route offers opportunities for off-the-shelf programmes to be pragmatically moulded to local context, in real-time”, p.24 – the methods of how this was done are unclear. When was the collected data analysed, what changes were made, how were these changes decided, when were they implemented? The discussion mentions a “strong theoretical grounding of the programme”, but no details on underpinning theory are provided in the manuscript.
--	---

VERSION 1 – AUTHOR RESPONSE

Reviewer: 1

Reviewer Name: Dimitrios Koutoukidis

Institution and Country: University of Oxford

UK

Please state any competing interests or state ‘None declared’: None

Please leave your comments for the authors below

Abstract

Overall

This is a useful study showing the challenges of scaling up weight loss programmes in routine care. Thank you for this positive feedback.

Major comments

Given the 20% retention rate at 1-year and the lack of accounting for missing data, authors should be more cautious in the interpretation of the effectiveness of the programme and this should be reflected both in the abstract, results, and discussion. I suggest that the abstract is overall framed more towards “lessons learned” rather than effectiveness.

We acknowledge this and agree that we should be more cautious in our interpretation of effectiveness. The abstract, results and discussion have all been revised, accordingly.

Methods

Page 11, line 16: it is unclear what does viable data mean. Is these complete cases? If so, why were the missing data ignored and what might be the implication of this? Although it is perfectly reasonable to report on completers, the resulting effect sizes are likely to be largely inflated by those who mostly adhered to the programme. Authors should also report effects on weight (and other measures) using multiple imputation (or full information maximum likelihood) at 12-weeks, given the very high dropout rate (which is typical in weight loss programmes).

We appreciate the reviewer's advice regarding our data and have amended phrasing to indicate analyses with complete cases. We have considered different imputation methods and have sought further advice from a statistician. We agree that our missing data is large, e.g. for follow-up we have only 20%. We consider though [as also noted by the reviewers] that this is an important finding in itself, in terms of understanding program engagement – and not unusual for weight management interventions. Jakobsen et al. (2017) suggest imputation where missing data are fewer than 40% <https://bmcmmedresmethodol.biomedcentral.com/articles/10.1186/s12874-017-0442-1>

The proportion of 'complete' cases in our evaluation is smaller than that which would need to be imputed. Furthermore, we consider that our missing data may not be completely at random, since those who do not return for follow-up are likely those who have not lost weight, i.e. disengaged with the programme. Instead, we have, as suggested, amended our interpretation of findings and present 'preliminary' outcome data as being hypothesis generating/exploratory.

Results

Authors should consider evaluating adherence through session attendance rather than attendance at 12-week follow-up. The latter evaluates retention rather than adherence. Please revise accordingly.

Thank you for these suggestions. Table 2 now includes evaluation of adherence through session attendance, and we now refer to 'attendance at 12-week follow-up' as 'retention'.

Minor comments

Please add the weight data for fitness only

Data for 'fitness-only' has been added for all variables in tables 3 and 4. We present descriptive data, and have not conducted inferential analyses due to the low $n = 3$.

Please rephrase 'persistence of weight loss' to 'weight loss maintenance' and provide numerical data for all 3 conditions.

We have rephrased 'persistence of weight loss' to 'weight loss maintenance' as suggested, and data are provided for all three conditions in table 3.

Line 7: Correct grammatical error
'is' has been added to the sentence

Line 8: Change "a" to lower case
This has been changed

Introduction

Line 12: Correct error "efficiencies"
We have clarified what is meant by 'efficiencies'.

Methods

Line 7 and throughout the manuscript: Please use person-first language when referring to people with overweight/obesity.

Thank you for raising this point – we have made amendments to ensure use of people-first language throughout the paper.

Line 8: unclear what "doubled" means
Reference to 'doubled' has now been removed

Line 9: delete "a"
Deleted

Page 8, line 5: country or county?

We do refer here to country, specifically in context of the English indices of multiple deprivation which describes lower super output areas according to their ranking in England.

Page 8, line 17: Please expand on how the allocation was implemented. Was it random, did people have a choice, etc.

We provide this information in our response to reviewer 2, below.

Page 11, line 18: Could the authors elaborate on the manuscript on why two different models were chosen instead of one model for all time points?

We have clarified this in the analysis section of the methods. Using complete cases, Kruskal-Wallis H tests were used to determine between-group differences at baseline and at 12 weeks and Wilcoxon-signed rank tests examined repeated measures differences between baseline and 12-week scores. We were unable to conduct a Factorial ANOVA, as might have been desirable, due to data not being normally distributed. Complete cases available at 52 weeks (n = 37) were considered similarly, but via separate analyses due to limited available data across the comparison groups.

Page 21, line 4: does this refer to the upper BMI threshold?

Yes, this is correct. We have clarified this by providing the BMI upper and lower thresholds in the text.

Discussion

Page 22, line 3: not sure where in the manuscript the 5% wl is reported? Missing data should be accounted for before comparison with other programmes can be made.

We have removed text comparing to commercial weight loss programme. We have amended the text to reflect 4% weight loss (our error and simply calculated from baseline and 52-week weight (kg) data in table 3) and cautioned that these findings are potentially hypothesis-generating, rather than an indication of effectiveness per se.

The authors argue that the “the prototyping evaluation format allowed for changes following programme commencement”. These changes would be really interesting and useful for the readers and I suggest you incorporate them in the results.

Thank you for this suggestion. We agree and have amended the text (also in response to reviewer 2) in methods, results and discussion to provide information on the refinements made. Briefly, these were (taken from the results section):

Real-time advice from Commissioners was considered during early stages of implementation, regarding the nature of comparison offers and thus design and outcome measurement. To better-target recruitment and change the process of engagement at referral point, entry criteria were altered (BMI ≥ 30 kg/m²) mid-way through the process. Implementation of the service offer on-site was adapted in response to delivery staff feedback: increased resource was made available, for example additional staffing to support delivery for the first wave of referrals, as well as capacity given for additional Momenta session preparation and extension of session delivery time. Follow-up activities (i.e., text or telephone contact) were implemented by staff during the process, to encourage adherence.

Key points should be less abstract and more close to the data.

We have clarified the aim and three objectives of the evaluation, and re-worked elements of the discussion to align with these. We think that this now presents a more balanced discussion of our data and still allows for consideration of the prototyping process within public health.

Reviewer: 2

Reviewer Name: Stephan Dombrowski

Institution and Country: University of New Brunswick, Canada

Please state any competing interests or state 'None declared': None declared

Please leave your comments for the authors below

Thank you for the opportunity to review this manuscript. It presents an interesting study examining the implementation of an “off the shelf” weight management program within a local context using a mixed methods design. Although the study examines a pressing issue in obesity and implementation, there are several comments that could be made on the current version of the manuscript.

There appears to be a disconnect between the objectives of the manuscript and large portions of the presented results. This is a study that is presented as examining prototyping, but the results focus primarily on what are probably underpowered weight loss and other secondary outcomes.

The current study is unlikely to be powered to provide an evaluation of effectiveness, the title might need amending and some of the presented results need to be interpreted with caution.

Thank you for these helpful comments. We have considered and addressed this disconnect through revisiting the manuscript aim and objectives and ensuring that a balance is presented (throughout the paper) for our three objectives (preliminary programme effectiveness, programme implementation, and exploring feasibility of prototyping for this kind of public health evaluation). We have amended the title and revised our interpretation of results throughout the manuscript as suggested. The discussion and conclusion have also been re-worked

Momenta is presented as “evidence based” throughout the manuscript, but it is unclear if this programme has been developed based on evidence, or if there is evidence of its effectiveness in inducing weight change? Please clarify.

Inserted on page 7: Momenta is an outcome-driven behavioural programme incorporating evidence-based behaviour change techniques that is designed to be delivered by fitness professionals in a leisure environment

Statements such as “Engagement of service users is encouraged”, p.5 need referencing – encouraged by whom?

We have amended this sentence to refer to the MRC guidance for developing and evaluating complex interventions, and cited this source.

The cost argument against user engagement in the introduction needs to be supported by evidence. It is not entirely clear why this would be relevant to make the case for the current study. Such statements might just antagonise some readers.

Thank you for noting this. We have re-worked the sentence to encompass a broader definition of cost, and cited evidence in support of this claim.

More detail on what is specifically meant by ‘prototyping’ would be useful and a review of background evidence would add value in the introduction. How would this differ from feasibility and pilot studies? There might be other study designs (e.g. open pilots) which are not called ‘prototyping’, but are similar to what has been attempted here could be examined to provide background information.

Additional information has been provided in the introduction to better-describe what is meant by ‘prototyping’. It is true that framing as a prototyping evaluation encompasses elements of other designs, however we feel that these in isolation do not fully describe the processes employed in our evaluation. For example, we agree that pilot, or feasibility studies have similarities to the design used in our work. A critical difference is that because we undertook an evaluation, we explored an existing scheme and thus the exchange of information was two-way between the evaluation team and the

public health team throughout the implementation process. We were able to communicate freely and adapt the implementation of the programme in 'real time', as well as provide a series of recommendations at the end, as would a pilot or feasibility study. We have also clarified how this worked in practice, as recommended in other comments.

How is the adaptability of the programme (mentioned on p. 6) determined in the context of the current study?

A number of additions have now been made to the manuscript regarding how prototyping allowed for adaptations to occur in real-time, and what those were. We have also amended text on page 6 to clarify this.

Please consider ending the introduction section with a clear statement of aim and objectives. We have amended the end of the introduction to clearly state the aim and objectives for the evaluation.

"Tier three weight management service" (p.7) might not be universally understandable to readers. We have removed 'tier three' and replaced with specialist weight management service.

More details on how the "prototyping approach" was determined and implemented would add value. More information on the specific design choices would be useful. The following has now been added to page 8:

Commissioners and providers had ideas about alternative delivery options and due to an established academic relationship, asked the study team for advice about robust evaluation that would allow for feedback in real time and at the end of the pilot.

How were service users allocated to the three groups (p.8)? If not at random this needs to be clearly stated.

We have added information about this into the manuscript (page 9). Participants were allocated into groups in order of receipt of referral forms (e.g. first form Momenta plus fitness, second form Momenta, third form fitness only, and so on in a repeating manner.) The provider then contacted participants by telephone to arrange attendance. If a participant was unable to attend the allocated group, (e.g. due to inconvenient session times) provider allocated them to a different group after discussion.

The number of study staff attending study meetings needs a denominator (p.8), or present as percentage to preserve anonymity.

These details have been added to page 8.

The number of HCPs and practice managers need to be added to the methods, p.10.

This information has been added to page 11.

Please provide details on how exactly anthropometric outcomes were collected.

Details have been included on page 10.

Please provide a brief description of the content of the interventions groups.

The following information has been added on page 7: Momenta sessions explored topics using interactive and experiential learning techniques including brainstorming, group activities and discussion, quizzes and games. At the end of each session, participants set goals focusing on one of the 12 key behaviours. At the beginning of each session, the group discussed the previous weeks' goals by exchanging stories and brainstorming challenges.

The patient and public involvement section states that “Commissioners, deliverers and service users were involved in the iterative evaluation.” But weren’t these subject of the investigation, rather than involved in the research study? Please clarify and provide more details if I misinterpreted this. We have now clarified this statement (mandatory for the journal) in the text, as follows: “Data from deliverers and service users, along with direct input from commissioners, fed into the iterative evaluation”.

The results focus on outcomes mainly, when the aim of the article seems to be to examine implementation using the prototyping approach. There seems to be a disconnect between the general introduction, and the presented results. Please see our previous response.

More emphasis should be given to dropout, which is high, as is not uncommon in these types of studies and programmes. Given the small sample size, no imputation methods being used, and the lack of randomisation (I assume), the interpretation of the findings should be more cautious. Thank you. We have amended the manuscript throughout (in response to suggestions from both reviewers) to reflect more cautious interpretation of quantitative findings.

One could argue that outcomes should be presented in line with recommendations for the CONSORT extension for pilot and feasibility trials (which this study resembles), e.g. removing p-values. It might be beneficial if the entire manuscript followed the CONSORT checklist, see <http://www.consort-statement.org/extensions/overview/pilotandfeasibility>. We thank the reviewer for this suggestion, which we have considered carefully. We certainly acknowledge the importance of the CONSORT extension for reporting pilot and feasibility trials, however we are reluctant to amend our framework to reflect these guidance because our work is not a randomised trial but rather a mixed methods service evaluation. We have, however, followed COREQ (Consolidated criteria for reporting qualitative research) guidelines for the qualitative data within this evaluation.

“the prototyping evaluation format allowed for changes following programme commencement, suggesting that this route offers opportunities for off-the-shelf programmes to be pragmatically moulded to local context, in real-time”, p.24 – the methods of how this was done are unclear. When was the collected data analysed, what changes were made, how were these changes decided, when were they implemented? Thank you, these points are very helpful. Please see our response to reviewer 1 for details of changes made throughout the manuscript.

The discussion mentions a “strong theoretical grounding of the programme”, but no details on underpinning theory are provided in the manuscript. We have amended our description of the programme (methods) in response to similar comments from both reviewers.

VERSION 2 – REVIEW

REVIEWER	Dimitrios Koutoukidis University of Oxford
REVIEW RETURNED	22-Jul-2019
GENERAL COMMENTS	Thank you for addressing our comments. Please see below some further suggested edits that can hopefully strengthen the manuscript.

	Abstract P46, Lines 20-21: It is unclear by just reading the abstract whether the reported weight loss is a between-group or within-group difference. Please clarify and also add weight data for the fitness only group. It would be worth adding a statement that there was no difference between-group at 12 weeks. P47, line 2: I would suggest delete the “remained at 52-weeks (p<0.05)”, giving the missing data and its exploratory nature. It is worth adding the 1-year retention rate in the abstract, perhaps at the point where you mention the long-term weight maintenance data. Strength and limitations I suggest the rate of missing data are incorporated in this section. Methods P51, line 13: It is unclear what ‘outcome-driven programme’ means. Thank you for adding more information about the programme. It would be useful to add a supplementary table on the structure of each session and the exact covered each week. Results Table 1: You may consider that Table 1 only reports on the 153 participants who attended the baseline assessment. It is also unclear why if 153 people attended the baseline assessment and were measured, there are only 150 people with BMI recorded, 123 with employment data, and 127 with education data. Were these measures introduced later on? If so, it would be worth adding a note about it at the bottom of the table. Table 2: The caption at the bottom is slightly confusing. I suggest rewording the explanatory text using the following structure: “% of participants who attended the 12-week assessment out of those who attended the baseline assessment” P60: I suggest that a line is added in the text that there were no between-group differences (I assume?) for the mental/depression/anxiety outcomes. I would be cautious to interpret a 1-point change in score as functionally and clinically meaningful and suggest this sentence is deleted unless you can provide a reference that such a change is in fact meaningful. The fact that the score dropped from the moderate category to the non-symptomatic category is simply due to the fact that the baseline value is very close to the cut-off. Discussion It would be worth discussing how long-term retention rates can be improved in future similar studies.
--	--

	It would be useful to add the COREQ checklist as part of the supplement.
--	--

REVIEWER	Stephan Dombrowski University of New Brunswick, Canada
REVIEW RETURNED	12-Jun-2019

GENERAL COMMENTS	I appreciate the authors addressing the majority of my concerns raised in the original review, and their time in providing comprehensive responses. I have a few more comments to offer. My central concern about over-stating the weight change results (as well as the other outcomes) of this study remain to a certain extent, although I think the authors made some useful changes towards toning this aspect of the study down. The weight findings are limited by the design of the study, the substantial dropout and the lack of accounting for dropout in the analysis, e.g. no baseline observation carried forward (BOCF) data is presented as often the case in weight management studies. Yet, these findings continue to feature prominently in the abstract and the interpretation of the study. I remain slightly concerned that the aim of the current study is to examine preliminary effectiveness. I think the study design is more suited to serve the other aims of explore local implementation, and consider feasibility of the iterative prototyping evaluation framework. The article summary section does not mention any of the preliminary effectiveness findings, and I think the findings highlighted in this section nicely represent what this study can tell us. The abstract states the first outcome in the primary and secondary outcomes measures to be weight loss followed by other weight related outcomes. I think it would be more useful to consider presenting the prototyping and implementation findings first (in abstract and manuscript), rather than weight and other outcomes given the substantial limitations of these measures. It might be better to include these outcome data for information and general description, rather than present them first and as indicators of effectiveness, and focus on recruitment, retention and data collection, rather than differences. I understand that the authors did not follow the consort extension for pilot and feasibility trials as suggested as they deem their study a mixed methods service evaluation. However, this study still includes elements of piloting and feasibility testing the off-the-shelf intervention, and as such the guidance should be considered in parts. In particular, I think the authors might want to consider removing p-values from the manuscript as the study was neither powered to examine differences, nor does the design allow meaningful inferences based on these comparisons. A more descriptive approach would be favourable. I understand that the PPI section is mandatory. However, the included sentence is not in line with common definitions of what constitutes PPI (e.g. INVOLVE defines public involvement in research as research being carried out 'with' or 'by' members of
---

	the public rather than 'to', 'about' or 'for' them. This includes, for example, working with research funders to prioritise research, offering advice as members of a project steering group, commenting on and developing research materials, undertaking interviews with research participants. – https://www.invo.org.uk/) It might be more useful to state that the current study did not involve PPI? Table 2, please include timepoint in the header.
--	--

VERSION 2 – AUTHOR RESPONSE

Reviewer(s)' Comments to Author:

Reviewer: 2

Reviewer Name: Stephan Dombrowski

Institution and Country: University of New Brunswick, Canada

Please state any competing interests or state 'None declared': None

Please leave your comments for the authors below

I appreciate the authors addressing the majority of my concerns raised in the original review, and their time in providing comprehensive responses. I have a few more comments to offer. My central concern about over-stating the weight change results (as well as the other outcomes) of this study remain to a certain extent, although I think the authors made some useful changes towards toning this aspect of the study down.

Thank you for taking the time to review our manuscript once again and for this positive response. We have further toned down our commentary about weight change and provide further information later in our responses.

The weight findings are limited by the design of the study, the substantial dropout and the lack of accounting for dropout in the analysis, e.g. no baseline observation carried forward (BOCF) data is presented as often the case in weight management studies. Yet, these findings continue to feature prominently in the abstract and the interpretation of the study.

We have amended the manuscript in respect of this point. Specifically, we have removed data points from the abstract and instead suggested potential for weight change as an associated outcome domain of interest. We have retained uptake and retention data in the abstract as we consider this to be a more important message. We have also toned down interpretation of outcomes throughout the manuscript.

I remain slightly concerned that the aim of the current study is to examine preliminary effectiveness. I think the study design is more suited to serve the other aims of explore local implementation, and consider feasibility of the iterative prototyping evaluation framework. The article summary section does not mention any of the preliminary effectiveness findings, and I think the findings highlighted in this section nicely represent what this study can tell us.

Thank you for making this point. We agree that the implementation aspect is the overall key driver for this evaluation. We have amended the objectives to tone down the focus on effectiveness. See page 6/7. "The main aim was to explore implementation of an 'off-theshelf' weight management

programme, Momenta24, in a challenging context. Specific objectives were to explore local implementation, consider feasibility of the iterative prototyping evaluation framework and identify preliminary programme outcomes.”

The abstract states the first outcome in the primary and secondary outcomes measures to be weight loss followed by other weight related outcomes. I think it would be more useful to consider presenting the prototyping and implementation findings first (in abstract and manuscript), rather than weight and other outcomes given the substantial limitations of these measures. It might be better to include these outcome data for information and general description, rather than present them first and as indicators of effectiveness, and focus on recruitment, retention and data collection, rather than differences.

Thank you for making this suggestion. We have removed reference to ‘outcome effectiveness’ in the subheadings and reframed the paper such that qualitative implementation is considered first, with quantitative data presented as secondary. We have not coloured all of this text in the manuscript due to the substantial movement of text but trust that this change is obvious when reading the revised document. We also note (in red text and prior to table 3) that these data are presented for information and general description.

I understand that the authors did not follow the consort extension for pilot and feasibility trials as suggested as they deem their study a mixed methods service evaluation. However, this study still includes elements of piloting and feasibility testing the off-the-shelf intervention, and as such the guidance should be considered in parts. In particular, I think the authors might want to consider removing p-values from the manuscript as the study was neither powered to examine differences, nor does the design allow meaningful inferences based on these comparisons. A more descriptive approach would be favourable.

Thank you for this helpful comment. We have debated further as a research team, and made some changes to overall manuscript as a result. For example we refer now to ‘preliminary outcome domains’ throughout (example in results section header) and have removed a lot of the inferences made from these results to provide a more descriptive account of findings. We have removed p values and reference to data values in the abstract, however have opted to retain p values in the results section, as useful indicators for the reader. Although the quantitative analysis is underpowered, there were still some differences observed which we think will likely be of interest to the reader. We hope that, along with the overall re-formatting of the quantitative elements in the manuscript, this is now satisfactory in presenting our data as preliminary.

I understand that the PPI section is mandatory. However, the included sentence is not in line with common definitions of what constitutes PPI (e.g. INVOLVE defines public involvement in research as research being carried out ‘with’ or ‘by’ members of the public rather than ‘to’, ‘about’ or ‘for’ them. This includes, for example, working with research funders to prioritise research, offering advice as members of a project steering group, commenting on and developing research materials, undertaking interviews with research participants. – <https://www.invo.org.uk/>) It might be more useful to state that the current study did not involve PPI?

This statement has been amended to reflect that the study did not involve any PPI.

Table 2, please include timepoint in the header.
Timepoints have now been included.

Reviewer: 1

Reviewer Name: Dimitrios Koutoukidis

Institution and Country: University of Oxford

Please state any competing interests or state ‘None declared’: None declared

Please leave your comments for the authors below

Thank you for addressing our comments. Please see below some further suggested edits that can hopefully strengthen the manuscript.

Thank you for taking the time to review our revised manuscript and suggesting these edits. We agree these strengthen the manuscript as detailed below.

Abstract

P46, Lines 20-21: It is unclear by just reading the abstract whether the reported weight loss is a between-group or within-group difference. Please clarify and also add weight data for the fitness only group. It would be worth adding a statement that there was no difference between-group at 12 weeks.

Within-group change has now been noted in the abstract. We have reduced the amount of information presented for quantitative data in the abstract, in light of suggestions from reviewer 1, thus this section no longer includes any specific weight data

P47, line 2: I would suggest delete the “remained at 52-weeks ($p < 0.05$)”, giving the missing data and its exploratory nature.

This has been deleted

It is worth adding the 1-year retention rate in the abstract, perhaps at the point where you mention the long-term weight maintenance data.

We have added the rate of data available at 52-week follow-up to the abstract.

Strength and limitations

I suggest the rate of missing data are incorporated in this section. We have amended point four to reflect this.

Methods

P51, line 13: It is unclear what ‘outcome-driven programme’ means. ‘outcome-driven’ has been removed from the text.

Thank you for adding more information about the programme. It would be useful to add a supplementary table on the structure of each session and the exact covered each week.

Thank you for this suggestion. A supplementary table has also now been included where we provide weekly session overview (negotiated with the program designers). We are unable to provide further detail due to copyright, however we think that this provides the necessary information for the manuscript.

Results

Table 1: You may consider that Table 1 only reports on the 153 participants who attended the baseline assessment. It is also unclear why if 153 people attended the baseline assessment and were measured, there are only 150 people with BMI recorded, 123 with employment data, and 127 with education data. Were these measures introduced later on? If so, it would be worth adding a note about it at the bottom of the table.

We have now included a footnote to table 1, explaining that data provided by different people (e.g. HCP on referral form, self-reported by participants during first session, leisure provider) and noting that missing data were either not followed up by leisure providers, or were presumed data entry errors by the evaluation team. We have also amended the title and values within the table to reflect ‘Demographic characteristics of referrals who started the programme’.

Table 2: The caption at the bottom is slightly confusing. I suggest rewording the explanatory text using the following structure: “% of participants who attended the 12-week assessment out of those who attended the baseline assessment”

Thank you for this, we have changed the caption as suggested and think it reads more clearly.

P60: I suggest that a line is added in the text that there were no between-group differences (I assume?) for the mental/depression/anxiety outcomes. I would be cautious to interpret a 1-point change in score as functionally and clinically meaningful and suggest this sentence is deleted unless you can provide a reference that such a change is in fact meaningful. The fact that the score dropped from the moderate category to the non-symptomatic category is simply due to the fact that the baseline value is very close to the cut-off.

We have added in a line to state no between-group differences were found. We have deleted the reference to the score dropping from moderate to non-symptomatic category as suggested. We have provided a reference for functional and clinically meaningful change but noted this to be small in the present evaluation.

Discussion

It would be worth discussing how long-term retention rates can be improved in future similar studies.

We have added the following paragraph to the discussion:

“We suggest that year-long (at least) follow-up be included and that this forms a key programme component from the outset. In other words, consideration might be given to any ‘light touch’ scheme support after 12 weeks and other means of obtaining follow-up data should be explored where service users disengage. Reasons for disengagement might also be usefully explored in future work.”

It would be useful to add the COREQ checklist as part of the supplement.

This has now been added. We have made a few minor amendments to the main text to emphasise adherence to COREQ guidance (noted in red text).

VERSION 3 – REVIEW

REVIEWER	Dimitrios Koutoukidis University of Oxford
REVIEW RETURNED	30-Aug-2019

GENERAL COMMENTS	Thank you for the revised manuscript. I believe that this round of revision has made the article much stronger. Some further minor points below for your consideration:  1. I would agree with reviewer 2 that p-values should be deleted and a descriptive presentation of the body of the results in pages 21-23 be favoured, especially for the 52-week data. The confidence intervals provide sufficient information for the reader to judge the precision and direction of a potential effect. 2. Abstract results, page 3, line 5: suggest change "programme end" to "12 weeks" to aid clarity. 3. Abstract conclusion, page 3, line 12: suggest change "has potential for" to "may have potential for". 4. page 21, line 8: consider changing "be functionally and clinically meaningful" to "approach to be functionally and clinically
--

	meaningful". Also this is a discussion point, so I suggest you move it down to the discussion section.
--	--

REVIEWER	Stephan Dombrowski University of New Brunswick, Canada
REVIEW RETURNED	16-Aug-2019

GENERAL COMMENTS	Thank you for again engaging with my comments on the manuscript. I am happy with the revisions made in response to my comments. Some final minor comments: The abstract should indicate that the potential effects are based on completers only. The presentation of 52-week retention should explicitly mention that 32%, 33% and 6% refers to retention, rather than dropout - this is not fully clear and could be easily misread. The conclusions should mention that preliminary weight effects are based on completers only.
---

VERSION 3 – AUTHOR RESPONSE

Reviewer: 2

Reviewer Name: Stephan Dombrowski

Institution and Country: University of New Brunswick, Canada

Please state any competing interests or state 'None declared': None declared

Please leave your comments for the authors below

Thank you for again engaging with my comments on the manuscript. I am happy with the revisions made in response to my comments.

*Thank you for your constructive support during the development of this final manuscript.

Some final minor comments:

The abstract should indicate that the potential effects are based on completers only.

*This has been now been indicated.

The presentation of 52-week retention should explicitly mention that 32%, 33% and 6% refers to retention, rather than dropout - this is not fully clear and could be easily misread.

*Retention has now been explicitly stated in the abstract.

The conclusions should mention that preliminary weight effects are based on completers only.

*Effects for completers only is now noted both in abstract conclusion and main article conclusion

Reviewer: 1

Reviewer Name: Dimitrios Koutoukidis

Institution and Country: University of Oxford

Please state any competing interests or state 'None declared': None declared

Please leave your comments for the authors below

Thank you for the revised manuscript. I believe that this round of revision has made the article much stronger.

*Thank you for your constructive support during the development of this final manuscript.

Some further minor points below for your consideration:

1. I would agree with reviewer 2 that p-values should be deleted and a descriptive presentation of the body of the results in pages 21-23 be favoured, especially for the 52-week data. The confidence intervals provide sufficient information for the reader to judge the precision and direction of a potential effect.

*P values have been removed from tables 3 and 4. Other small changes have been made to the Methods – data analyses, and Results on pages 20-23 to highlight descriptive nature of results.

2. Abstract results, page 3, line 5: suggest change "programme end" to "12 weeks" to aid clarity.

*Programme end has been amended to read '12 weeks'.

3. Abstract conclusion, page 3, line 12: suggest change "has potential for" to "may have potential for".

*Changed as suggested.

4. page 21, line 8: consider changing "be functionally and clinically meaningful" to "approach to be functionally and clinically meaningful". Also this is a discussion point, so I suggest you move it down to the discussion section.

*We have made this change and moved to the discussion section.